# Heavy Metals, Proximate Analysis and Brine Shrimp Lethality of *Vernonia amygdalina* and *Ocimum gratissimum* Growing in Crude Oil-Rich Delta State, Nigeria

**DOI:** 10.3390/foods10122913

**Published:** 2021-11-24

**Authors:** Oluwatofunmilayo Arike Diyaolu, Alfred F. Attah, Emmanuel T. Oluwabusola, Jones Olanrewaju Moody, Marcel Jaspars, Rainer Ebel

**Affiliations:** 1Marine Biodiscovery Centre, Department of Chemistry, University of Aberdeen, Aberdeen AB24 3UE, UK; r01eto16@abdn.ac.uk (E.T.O.); m.jaspars@abdn.ac.uk (M.J.); r.ebel@abdn.ac.uk (R.E.); 2Department of Pharmacognosy and Drug Development, Faculty of Pharmaceutical Sciences, University of Ilorin, Ilorin 200132, Nigeria; attah.fau@unilorin.edu.ng; 3Department of Pharmacognosy, Faculty of Pharmacy, University of Ibadan, Ibadan 200132, Nigeria; jo.moody@mail.ui.edu.ng

**Keywords:** proximate analysis, brine shrimp assay, crude oil exploration, atomic absorption spectrometry

## Abstract

*Vernonia amygdalina* (VA) and *Ocimum gratissimum* (OG) are among the most frequently consumed vegetables in Kokori and Abraka communities of Delta State, Nigeria. However, the continuous crude oil exploration and spillages in Kokori may threaten their safety for use as food and medicine. Twelve samples of VA and OG obtained from crude oil-rich and crude oil-free communities were comparatively analysed for proximate composition, heavy metals, and cytotoxicity. Data obtained were subjected to various multivariate statistical techniques, including principal component analysis (PCA), biplot, and analysis of variance (ANOVA), to investigate the correlations between the vegetables from the different communities and the effect of crude oil exploration and spill on plant biomass. Results obtained indicate a significant difference (*p* < 0.05) in the proximate composition of VA and OG and higher heavy metal content for VA from the crude oil-spill Kokori. Two VA collections from Kokori were exceptionally toxic to cellular crustaceans.

## 1. Introduction

The predominantly endemic tropical vegetable, *Vernonia amygdalina* Del. (Asteraceae), also known as the bitter leaf, is a staple in the Nigerian diet that has been domesticated as an essential source of nutrients and a phytomedicine [1]. *Vernonia amygdalina* (VA) is a commonly cultivated tropical shrub that grows about 2–5 m tall. Its pubescent branches become glabrous as they reach maturity. The dark green leaves are displayed alternately on the branches, measure between 4–6 mm in diameter, and are ovate-shaped, entire, or finely toothed with a characteristic odour and a bitter taste [2]. The leaves are consumed as food, vegetables, or culinary spice in soups. *Ocimum gratissimum* L. (Lamiaceae) is an aromatic ethnomedicinal plant native to Tropical Africa and Asia, but has been naturalised in many parts of sub-tropical geographies [3]. In Nigeria, *Ocimum gratissimum* (OG) leaves, also popularly known as African basil, are eaten in soups and salads. Due to its characteristic aromatic smells used as a spice and food flavouring in many local Nigerian dishes, the leaf oil is also used as a food preservative and an anti-infective and anti-inflammatory agent [4]. It grows as an erect shrub, with a maximum height of 2 m. Both VA and OG are valuable healthy and nutritious vegetables, low in calories and high in proteins, vitamins, iron, zinc, selenium, fibres, and minerals [5].

Furthermore, they have medicinal uses, particularly among the indigenous women in Delta State, Nigeria. The leaves are macerated in an aqueous medium, and the extracted juice is administered during the last trimester of pregnancy to stimulate uterine contractions [6]. Although VA and OG thrive on all soil types, they grow better on humus-rich soils [7]. However, the crude oil exploration and petrochemical effluent release into Nigeria’s oil-producing coastal regions threaten the growth and survival of these plants. Kokori community is a typical example of such communities in Delta State where crude oil exploration and effluent release is common. Kokori has over fifty oil wells, flow stations, and oil pipelines; it is reported to produce the second-best crude oil globally due to its low sulphur content [8]. The ongoing crude oil exploration in Kokori causes crude oil spills and has led to the death of many plants. However, *VA* and *OG* survive in this harsh environment. Crude oil spills are a source of heavy metal pollution of water bodies and terrestrial habitats; they are a complex blend of hydrocarbons, non-hydrocarbon compounds, and heavy metals found in subsurface deposits worldwide [9]. Heavy metals may be concentrated in leafy vegetables’ edible and non-edible parts [10]. Prolonged consumption of such vegetables containing an unsafe concentration of heavy metals may lead to chronic accumulation of heavy metals in vital organs such as the kidney and liver of human beings and animals, which may disrupt various biochemical processes, leading to gastrointestinal, cardiovascular, renal, bone, and nervous diseases [11].

Nigeria produces a million barrels of crude oil daily, making it Africa’s topmost crude oil producer and ranking 11th globally. Among Nigeria’s eight crude oil-producing states, Delta represents the second-largest oil-producing state, contributing considerably to the daily crude oil output [10].

Until now, the effect of crude oil spills on the nutritional value and safety profile of *VA* and *OG* used by the local communities has not been investigated. Nevertheless, the residents continue to consume them. Not all communities in Delta state are crude oil-rich, which suggests that only oil-rich communities have crude oil exploration and exploitation activities taking place among them. In the Delta state, an example of a crude oil-free community is Abraka. The present study compared the basic composition, heavy metal content (chromium, cadmium, lead, mercury, arsenic and thallium), and *VA* and *OG* safety profile in Kokori crude oil-rich and Abraka crude oil-free communities using standard analytical methods, atomic absorption spectroscopy (AAS), and brine shrimp toxicity assay.

## 2. Materials and Methods

### 2.1. Collection and Sample Preparation

The vegetables (*Vernonia amygdalina* and *Ocimum gratissimum*) were collected in March 2021 from six different sites, three within crude oil-rich Kokori and three within the crude oil-free Abraka community (Table 1). Both communities are located within Ethiope east local government area, Delta State, Nigeria (Figure 1). The vegetables were authenticated at the Forest Herbarium Ibadan (FHI), Nigeria, where voucher specimens for *Vernonia amygdalina* (FHI 113102) and *Ocimum gratissimum* (FHI 113103) were deposited. The nutritional and heavy metal analysis experiments were conducted at the Marine Biodiscovery Centre, Department of Chemistry, University of Aberdeen, Aberdeen and the Department of Human Nutrition and Dietetics, College of Medicine, University of Ibadan, Nigeria. The brine shrimp toxicity assay was conducted at the Pharmacognosy and Drug Development Department, University of Ilorin, Nigeria. The vegetables (leaves) were cleaned without washing, transported to the laboratory within 4 h of collection, and placed temporarily in glass vessels at −4 °C. Samples were assayed individually.

### 2.2. Analytical Methods

#### 2.2.1. Moisture

In order to obtain moisture contents, vegetable samples were dried in a moisture oven at 105 °C overnight for 17 h [13].

#### 2.2.2. Crude Proteins

The vegetable’s crude proteins were evaluated using the macro Kjeldhal method [14]. The samples were digested in concentrated sulphuric acid in a Kjeltec digestion apparatus (1007 Digestion Unit, Tecator, Sweden). The digested samples were then distilled after the addition of alkali. Ammonia was given off and collected in 4% boric acid in the Kjeltec Automatic Distilling Unit. The resulting boric acid contains the ammonia released from the digested samples; it was then titrated against 0.1 N HCl. The final crude protein was calculated by multiplying the nitrogen content obtained by a factor of 6.25.

#### 2.2.3. Crude Fat

A Soxhlet apparatus was used to analyse the fat content present in the vegetable samples; the solvent was petroleum ether.

#### 2.2.4. Ash

The ash values were obtained by weighing the samples before and after igniting at 500 °C for 24 h.

#### 2.2.5. Total Carbohydrates

The total carbohydrates present in the vegetables were determined using the formula [14]:Total carbohydrates (g ÷ 100 g fresh weight) = 100 − (% moisture) − protein content (% fresh weight) − crude fat (% fresh weight) − ash (% fresh weight)(1)

#### 2.2.6. Comparative Analysis of Experimental Data to Reference Material Data

The analytical methods’ accuracy, sensitivity, and reliability described above were validated using the certified reference material (CRM) IRMM-411 (Corn flour) supplied by Merck.

The CRM was preserved by storing it under strictly controlled conditions. The measurements conducted on the CRM were done in triplicates, and data obtained were compared with certified values. The Ash, protein, and fat values in the CRM were 0.20 ± 0.03, 1.75 ± 0.02, and 1.2 ± 0.10. The analysis of IRMM-411 using the above experimental methods gave values of 0.20 ± 0.01, 1.73 ± 0.01 and 1.1 ± 0.10, respectively. Thus, our analysis is in excellent agreement with the certified values.

#### 2.2.7. Heavy Metals

The vegetable samples were dried for 24 h at 105 °C. Following complete dehydration, the samples (2 g each) were digested with 8 mL of concentrated nitric acid (HNO_3_), 2 mL concentrated sulphuric acid (H_2_SO_4_), and 2 mL hydrogen peroxide (H_2_O_2_) and heated for 4 h at 70 °C; then, it was cooled; 20 mL of distilled water were included, and further digestion was carried out by heating with concentrated HNO_3_ and H_2_SO_4_.

Subsequently, drops of concentrated HNO_3_ were added until total oxidation of the organic matter, assumed to be achieved once no further darkening of the solution was seen, and on continual heating, a clear yellow solution was eventually achieved. The mixture was cooled, and any insoluble solids remaining was filtered with Whatman No. 42 filter paper and then transferred to suitable containers. The pH of the solutions was determined and used for spectrophotometric analysis.

The amount of chromium, cadmium, lead, mercury, arsenic, and thallium in the vegetable samples were assessed with an HGA graphite furnace, and argon was utilised as the inert gas. The wavelengths (nm) used to determine chromium, cadmium, lead, mercury, arsenic, and thallium were: 358, 229, 283, 254, 194, and 364, respectively. Standards used to create calibration curves were purchased from Datolab Chemicals, Ibadan, Nigeria. Three calibration points were used for each metal.

The heavy metal measurements were validated by analysing ERM-CE 278, the standard reference for this experiment (Table 2). The data obtained agree with the certified values. For reliability and accuracy of the measurements, the digested reference material solution was included in the loading list and analysed after every ten samples ran. The calibration curve was reconstructed each time there was a deviation of more than 10%. The table also shows the heavy metals permissible limits in vegetables set by the FAO/WHO and the US FDA [15].

### 2.3. Brine Shrimp Toxicity Assay

The vegetable samples were screened for toxicity using larvae (nauplii) of *Artemia salina* (brine shrimp) [16]. The samples were reconstituted in saltwater (35 g/1 L of water) for brine shrimp lethality assay. The test was performed in triplicate using calibrated 15 mL Eppendorf tubes, with extract concentrations of 1000, 500, 100, 10, and 1 µg/mL. Incubation was carried out in natural seawater for 36 h. Following this, the nauplii, at least ten each, were aspirated and carefully dispensed in the vials containing the test samples. Incubation at room temperature under light illumination for another 24 h was conducted, and the surviving nauplii in each well were determined using a hand lens. Seawater was used as the negative control without the test substance, while potassium dichromate was the positive control. Determination of the 50% lethal concentrations (LC_50_) of the vegetable test samples was conducted using Finney’s probit analysis. The mean ± standard deviation of the mean LC_50_ was calculated from three independent experiments. Dilutions of test samples that did not show toxicity were considered non-toxic [17].

### 2.4. Statistical Analysis

All analyses were carried out in triplicates, expressed as mean ± standard deviation (SD).

The proximate and heavy metal results obtained were analysed by one-way analysis of variance (ANOVA) using the Statistical Package for the Social Sciences, IBM SPSS statistics for windows, version 22 (IBM Corp., Armonk, NY, USA). Statistically significant differences were addressed by comparing the mean chemical values pair-wise according to the confidence intervals of 95% based on ANOVA’s combined standard deviation. The variations between vegetable samples were analysed using principal component analysis (PCA) and biplot with the MetaboAnalyst software, version 4.0, Xia lab, Mcgill, CA [18].

## 3. Results and Discussion

The toxic metals studied in these experiments include As, Cd, Cr, Hg, Pb, and Tl. Toxic metals such as cadmium, chromium, and mercury were not detected in the investigated vegetable samples despite crude oil exploration activities in Kokori. However, low levels of Arsenic and Thallium were detected while lead accumulation was higher for vegetables collected from Kokori (0.03 µg/g), which may be associated with the crude oil exploration in this region. Overall, these toxic metals are below the permissible limits set by the World Health Organisation [19] for heavy metal occurrence in plants. The shallow crude oil reservoir level may have reduced the potentiality of limiting exploration operations to the upper surface of the earth, where there is a minimum deposit of heavy metals. Meanwhile, lead has the potential to bioaccumulate in toxic levels in the future.

Oil spills on agricultural soils generally tend to diminish plant growth [20], although low oil contamination levels may also catalyse growth [21]. Contamination due to crude oil spillages and gas flaring causes environmental and soil characteristics changes, and many of these changes can be directly related to soil microbial activity. Several species of bacteria, yeasts, and fungi are known to attack hydrocarbons. These occur throughout temperate environmental conditions, and microbial activity increases with increased oil spillages. Microbial by-products may change soil moisture retention and release, thereby changing the moisture content of plants [22]. For instance, crude oil exploration in Iraq has been documented to differentially affect the growth and productivity of important plants (*Vicia faba* and *Recinous communis*) used as food and medicine [23]. When the nutritional value of VA and OG vegetables are evaluated, perhaps the most crucial parameter is their moisture content/dry matter, which directly affects the nutrient content of the vegetables [13]. In Table 3, the moisture content of the vegetables VA and OG, collected from the crude oil-spill environment (Kokori), ranged from 9.50–10.76% and 8.11–9.86%, respectively. On the other hand, the moisture content of VA and OG from Abraka (crude oil-free) ranged from 8.88–9.90% and 7.51–9.60%. Although they are both in line with earlier published data [24], the moisture content of vegetables from Kokori is significantly higher than that of vegetables from Abraka; this difference may be attributable to the crude oil exploration constantly taking place in Kokori, but not occurring in Abraka which may have resulted in a reduced stomatal conductance, (an estimate of the amount and diffusion rate of carbon dioxide (CO_2_) into leaves for photosynthesis) and water loss via transpiration. Plants growing in crude oil spill contaminated environments have been reported to show altered stomatal conductance and this is consistent with an experiment conducted by Odukoya and colleagues. They documented the impact of crude oil contamination on the stomatal conductance of leafy vegetables [25].

Crude fat in VA and OG includes several lipids, fatty acids, sterols, sterol esters, and mono, di, and triglycerides [5,26]. Fatty acids are involved in various physiological activities, including metabolism, and are indispensable substances in plants. The fat content in Kokori derived VA and OG ranged from 0.73–0.93% and 0.83–1.50%, respectively, compared to a higher fat content range of 1.10–1.70% and 1.38–2.17% for VA and OG from crude oil-free Abraka. Oil spills have a long-term effect on agricultural soils. The lower fat accumulation in vegetables from Kokori may be related to the stress induced by soil contamination of crude oil during exploration, altering the biosynthesis of fatty acids; this is in agreement with a study published by Liu et al., who documented the effect of crude oil spill stress on chlorophyll *a* and fatty acids [27]. Thus, exposure of the vegetables to petroleum products may likely harm fatty acid biosynthesis.

VA’s protein concentration sourced from Kokori and Abraka ranged from 22.99–23.71% and 23.71–28.81% consecutively; both ranges agree with published data that VA contained 17 to 33% crude protein [28,29]. However, VA from Abraka appears to accumulate higher protein levels, albeit within the documented range, suggestive of a possible absence of the effect of crude oil mining activities on protein levels in VA Interestingly, the crude proteins analysed in OG from Abraka ranged from 14.89–23.86% compared to 19.40–24.52% for OG from Kokori. These also agree with literature reports on observed higher crude protein levels in regions where crude oil exploration occurs against other crude oil-free regions in Nigeria [30].

The vegetables’ total carbohydrate content, calculated by difference, varied from 46.56–49.54% and 45.79–46.92% for VA collected from Abraka and Kokori. The crude oil spill bitter leaf vegetable showed a lower carbohydrate content. Interestingly, OG collected from crude oil-spill Kokori displayed a lower carbohydrate concentration of 46.83–56.16% than OG from Abraka, 46.92–64.72. The continuous crude oil spills in Kokori make the agricultural fields and vegetables exposed to petroleum hydrocarbons (PH), reducing plant growth. It is reported that petroleum hydrocarbon contamination causes an alteration in the carbon/nitrogen (C/N) ratio, nitrogen deficiency, and may also lead to stunted plant growth. Thus, the lower carbohydrate content seen in VA and OG from Kokori may be associated with a deceleration of the enzyme-catalysed steps involved in the biosynthetic pathway of carbohydrates and may be potentially linked to compounds released into the adjoining soil during crude oil mining activities [31].

Vegetables are good sources of minerals. The ash contents varied from 18.56–20.24% and 18.13–19.70% for VA collected from Kokori and Abraka, with VA from Kokori showing a higher ash content. The ash content for OG leaves from Kokori also showed higher values of 15.20–18.68% than OG from Abraka 11.21–18.11%, indicating a possible contamination resulting in a higher accumulation of inorganic materials [32]. This observation may have arisen from the crude oil mining activities in Kokori, which is not happening in Abraka. Since the heavy metal analysis findings revealed a low accumulation of heavy metals, these minerals may be other less toxic elements.

VA leaves from Kokori showed a higher fibre content of 19.10–22.91% compared to 15.60–18.21% for VA from Abraka, consistent with an earlier report from South-Eastern Nigeria. Additionally, the proximate analysis revealed that OG from Kokori has 24.45–30.11% higher fibre content than OG from Abraka (18.20–23.25%). Different ranges of values (4–20%) for crude fibres have been reported by various authors who investigated the fibre contents of vegetables during research periods that are far apart in years. However, values obtained from this study appear to be higher than previously reported in the study plants [33].

Metal concentrations in the studied twelve VA and OG vegetable samples (Table 4) were all below standard limits set by FAO/WHO [34]. These results are similar to literature values documented for the heavy metal analysis carried out on some fruits in crude oil-rich communities in Delta State [35]. Cd, Cr, and Hg concentrations were under the detection limit of the method used. The detection limits for Cd, Cr, and Hg are 0.02 µg/g for each element. From the above results, despite the crude oil exploration activities ongoing in Kokori, it can be seen that all collected samples have low levels of toxic metals.

In order to visualise the subtle pattern of similarities and variations among the vegetable samples with respect to their collection sites, Principal Component Analysis (PCA) was utilised. The multivariate data analysis performed on the proximate composition data (Table 3) revealed a significant separation between VA and OG samples collected from crude oil-free Abraka and crude oil-rich Kokori. The PCA score plot comprising three replicates of twelve *Vernonia amygdalina* and *Ocimum gratissimum* samples were acquired as shown in Figure 2A. Samples collected from crude oil-free Abraka; VA-OAB, VA-UAB, VA-AAB, OG-OAB, OG-UAB, and 0G-AAB were separated from samples collected from crude oil-rich Kokori; VA-EGK, VA-K, VA-SK, OG-EGK, OG-K, and OG-SK by PC1. However, the sample VA-EGK positively correlates with VA-OAB; this may be because of the closer proximity of Egbo Kokori community to crude oil-free Oria, Abraka. The first two components, PC1 and PC2, explained 83.3% and 12.0% of the total variance, respectively.

Furthermore, to elucidate specific relationships between proximate composition and the vegetable species, a biplot (Figure 2B) was generated; it gives a broader view of the effect of all chemical variables based on the results of the principal component analysis. Variables with longer arrows are more critical in producing effects, while those with the same direction show a positive correlation. Vegetable samples positioned close to an arrow of a variable show a strong relationship. Thus, high amounts of carbohydrates are seen in the leaves of *Ocimum gratissimum* collected from Oria, Abraka (OG-OAB), whereas *Ocimum gratissimum* collected from Egbo, Kokori (OG-EGK) is the richest in fibre. The *Vernonia amygdalina* samples collected from crude-oil rich Kokori (VA-SK and VA-K) are essential for their high Moisture, protein, and ash values. These three variables correlate positively.

The brine shrimp lethality assay results (Table 4) indicated that all *Ocimum gratissimum* samples collected from the crude oil-free Abraka and crude oil-rich Kokori have median lethal concentration (LC_50_) value greater than 1000 µg, suggesting the vegetables are non-toxic to tested organisms [16,36].

Additionally, this is in line with the low concentration or absence of toxic heavy metals reported in this study for OG (Appendix A). It also agrees with an earlier report on the absence of toxicity of the plant collected from Western Nigeria against nauplii of *Artemia salina* [37]; this may imply that OG has not been significantly affected by crude oil exploration in Kokori. *Vernonia amygdalina* samples presented high, moderate, and low toxicity profiles based on sample collection sites. Generally, VA samples were more toxic to *Artemia salina* than OG samples. However, two VA samples, VA-EGK and VA-K, collected from the crude oil-rich Kokori were toxic (LC_50_ of 111 µg) and highly toxic (LC_50_ of 9 µg) to the tested organisms. In particular, VA-K showed higher toxicity (LC_50_ of 9.23 µg) compared to the positive control (LC_50_ of 23.08 µg), potassium dichromate (Table 4). While documented reports have supported the moderate toxicity of VA [38,39]. VA-K showed an unusually high toxicity profile to tested organisms. Future research should therefore focus on investigating what might have led to the observed toxicity.

## 4. Conclusions

This study has shown that the vegetables *Vernonia amygdalina* and *Ocimum gratissimum* are good sources of nutrients and trace valuable minerals whose nutritional potentials could easily be compromised when growing in regions where crude oil exploration is taking place, as that seen in Kokori, Delta state. Based on the PCA done, there is an observed significant difference between the nutritional content of the vegetables collected from Abraka, the crude oil free community, and Kokori, the crude oil spill community. However, the observed low levels of toxic metals (Pb, Cd, and As) in VA and OG samples collected from the crude oil-rich Kokori shows that the crude oil pollution in Kokori has little to no effect on them. These documented bizarre low levels may be related to the shallowness of the crude oil wells in Kokori, which do not permit exploration activities to get to the deep part of the earth where heavy metals are abundant. However, *Vernonia amygdalina*, in particular, samples collected from the crude oil-rich Kokori, presented toxicity to brine shrimps and will warrant further toxicological evaluation for evidence-informed and safe consumption of the plants in the rural community.

## Figures and Tables

**Figure 1 foods-10-02913-f001:**
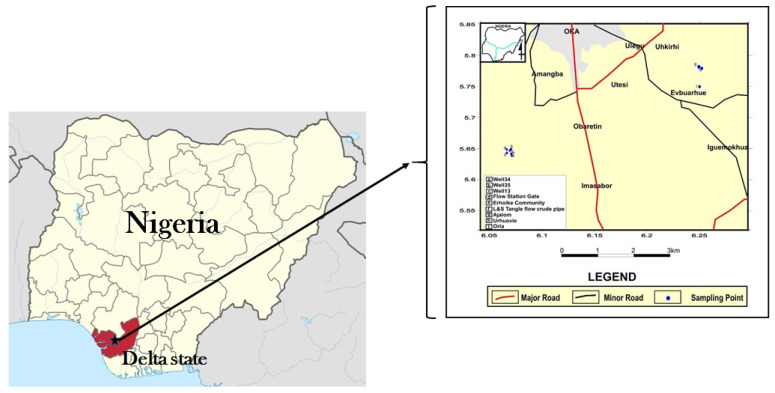
Map showing study area and collection sites in Delta State. Map of the study area has been adapted from Ozabor and Obaro, 2016 [12] while the Nigerian map was adapted from https://en.wikipedia.org/wiki/Delta State and www.deltastate.gov.ng (accessed on 21 September 2021).

**Figure 2 foods-10-02913-f002:**
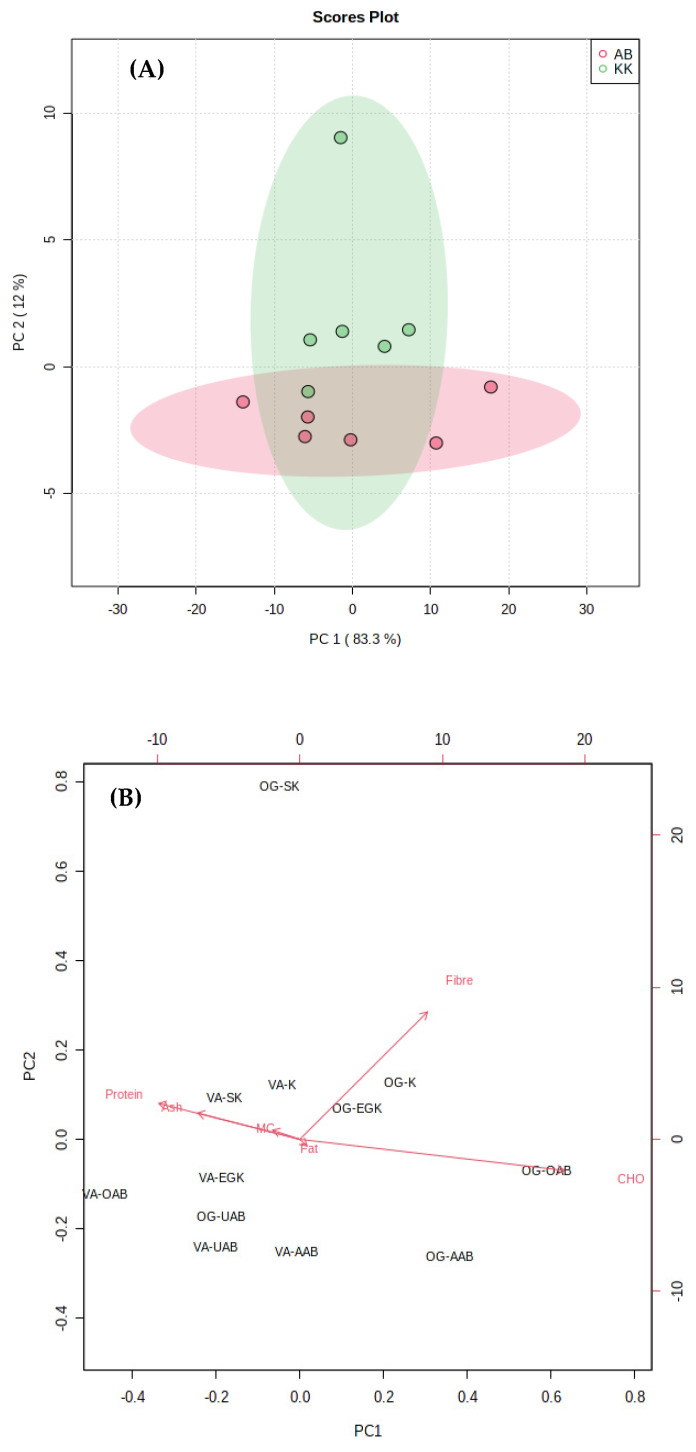
(**A**) PCA scores for the proximate composition of VA and OG collected from crude oil-spill Kokori and crude oil-free Abraka communities. Green circles correspond to vegetables from crude oil spill areas (Kokori). Pink circles correspond to vegetables from crude oil-free areas (Abraka). (**B**) Biplot based on principal component analysis of VA and OG chemical composition and species arrangement.

**Table 1 foods-10-02913-t001:** Zone, coordinates, and collection sites.

No.	Sample	Zone	Coordinates	Place of Collection	Codes
1	*Vernonia amygdalina*	Abraka	5°45′ N 6°15′ E	Oria	VA-OAB
2	*Ocimum gratissimum*				OG-OAB
3	*Vernonia amygdalina*	Abraka	5°47′ N 6°15′ E	Urhuovie	VA-UAB
4	*Ocimum gratissimum*				OG-UAB
5	*Vernonia amygdalina*	Abraka	5°47′ N 6°15′ E	Ajalom	VA-AAB
6	*Ocimum gratissimum*				OG-AAB
7	*Vernonia amygdalina*	Kokori	5°38′ N 6°04′ E	Egbo	VA-EGK
8	*Ocimum gratissimum*				OG-EGK
9	*Vernonia amygdalina*	Kokori	5°38′ N 6°15′ E	Kokori	VA-K
10	*Ocimum gratissimum*				OG-K
11	*Vernonia amygdalina*	Kokori	5°38′ N 6°22′ E	Samagidi	VA-SK
12	*Ocimum gratissimum*				OG-SK

**Table 2 foods-10-02913-t002:** Certified reference material and the determined values for the measured heavy metals.

Metals	Certified Value (µg/g)	Measured Value ^a^ (µg/g)	FAO/WHO (µg/g)	US FDA (µg/g)
As	6.05 ± 0.12	5.59 ± 0.19	10.00	10.00
Cd	0.36 ± 0.01	0.35 ± 0.01	0.20	0.20
Cr	0.75 ± 0.06	0.71 ± 0.04	1.30	1.30
Hg	0.25 ± 0.02	0.21 ± 0.01	1.00	1.00
Pb	2.00 ± 0.02	1.99 ± 0.01	10.00	10.00
Tl	5.25 ± 0.11	5.30 ± 0.13	0.20	0.20

^a^ An average of three digestions and triplicate measurements.

**Table 3 foods-10-02913-t003:** Proximate composition of the vegetables on a dry weight (d.w.) basis.

S/N	Samples’ ID	% Moisture Content	% Crude Protein	% Crude Fat	%Crude Fibre	% Ash	% CHO
1	VA-OAB	9.66 ± 0.02 ^a^	28.81 ± 0.05 ^a^	1.10 ± 0.02 ^a^	15.60 ± 0.03 ^a^	19.70 ± 0.52 ^a^	46.56 ± 0.10 ^a^
2	OG-OAB	7.51± 0.01 ^c^	14.89 ± 0.12 ^c^	1.68 ± 0.01 ^c^	28.00 ± 0.05 ^c^	11.21 ± 0.02 ^c^	64.72 ± 0.16 ^c^
3	VA-UAB	8.88 ± 0.03 ^d^	24.30 ± 0.12 ^d^	1.70 ± 0.02 ^c^	17.33 ± 0.03 ^d^	18.22 ± 0.02 ^d^	49.54 ± 0.09 ^b^
4	0G-UAB	9.60 ± 0.02 ^d,e^	23.86 ± 0.11 ^c,e^	1.38 ± 0.01 ^j,i^	18.2 ± 0.05 ^j^	18.11 ± 0.02 ^j^	46.92 ± 0.11 ^j^
5	VA-AAB	9.90 ± 0.02 ^i^	23.71 ± 0.12 ^b^	1.39 ± 0.01 ^h,i^	18.21 ± 0.07 ^i^	18.13 ± 0.05 ^d^	46.88 ± 0.09 ^b,d^
6	OG-AAB	8.32 ± 0.02 ^f^	17.21 ± 0.12 ^f^	2.17 ± 0.02 ^f^	23.35 ± 0.04 ^f^	12.48 ± 0.01 ^f^	59.80 ± 0.15 ^f^
7	VA-EGK	9.50 ± 0.02 ^b^	23.64 ± 0.12 ^b^	0.93 ± 0.02 ^b^	19.10 ± 0.05 ^b^	20.24 ± 0.02 ^b^	46.72 ± 0.07 ^b^
8	OG-EGK	9.33 ± 0.03 ^h^	20.58 ± 0.12 ^h^	1.30 ± 0.01 ^h^	24.45 ± 0.07 ^h^	15.20 ± 0.01 ^h^	53.59 ± 0.14 ^h^
9	VA-K	9.51 ± 0.03 ^b^	23.71 ± 0.13 ^b^	0.73 ± 0.05 ^e^	22.91 ± 0.09 ^e^	18.56 ± 0.05 ^e^	46.92 ± 0.09 ^e^
10	OG-K	8.11 ± 0.01 ^g^	19.40 ± 0.12 ^g^	0.83 ± 0.05 ^b,e,l^	26.13 ± 0.05 ^g^	15.48 ± 0.01 ^g^	56.16 ± 0.15 ^g^
11	VA-SK	10.76 ± 0.04 ^l^	22.99 ± 0.12 ^l^	0.83 ± 0.05 ^b,e,l^	21.17 ± 0.017 ^l^	19.44 ± 0.05 ^l^	45.79 ± 0.15 ^l^
12	OG-SK	9.86 ± 0.04 ^b^	24.52 ± 0.12 ^d^	1.50 ± 0.01 ^k,j^	30.11 ± 0.02 ^k^	18.68 ± 0.02 ^k^	46.83 ± 0.16 ^b,d,i^

VA—*Vernonia amygdalina*, OG—*Ocimum gratissimum*, OAB—Oria Abraka, UAB—Urhovie Abraka, AAB—Ajalom Abraka, EGK—Egbo Kokori, K—Kokori, SK—Samagidi Kokori. Each value is a mean of triplicates ± SD. Means with no common letters within a column significantly differ (*p* ≤ 0.05); *n* = 3.

**Table 4 foods-10-02913-t004:** Brine shrimp lethality of aqueous extracts of VA and OG vegetables.

S/N	Sample	Mortality (%)					95% CI	LC_50_ Value (µg)	#Inference
		Concentration (µg/mL)							
		1	10	100	500	1000			
1	VA-OAB	10.00	20.00	20.00	46.66	60.00	6.92 × 10^3^	7.74 × 10^2^	Mildly toxic
2	OG-OAB	3.33	10.00	16.55	30.00	36.66	9.46 × 10^5^	6.09 × 10^3^	Non-toxic
3	VA-UAB	6.66	3.33	20.00	30.00	46.66	3.64 × 10^5^	3.88 × 10^3^	Non-toxic
4	OG-UAB	3.33	6.66	3.33	36.66	33.30	2.50 × 10^5^	4.66 × 10^3^	Non-toxic
5	VA-AAB	0.00	0.00	3.33	100.00	100.00	0.00	6.14 × 10^3^	Non-toxic
6	OG-AAB	10.00	10.00	13.33	30.00	40.00	2.42 × 10^7^	1.45 × 10^4^	Non-toxic
7	VA-EGK	20.00	40.00	3.33	43.33	93.30	3.42 × 10^2^	1.12 × 10^2^	Toxic
8	OG-EGK	23.33	26.36	30.00	36.66	60.00	2.16 × 10^7^	1.67 × 10^3^	Non-toxic
9	VA-K	40.00	50.00	60.00	66.66	76.66	46.00	9.23	Highly toxic
10	OG-K	26.66	30.00	33.33	40.00	56.66	1.56 × 10^6^	1.21 × 10^3^	Non-toxic
11	VA-SK	10.00	16.66	20.00	20.00	43.33	1.15 × 10^7^	3.25 × 10^4^	Non-toxic
12	OG-SK	13.33	20.00	80.00	96.66	100.00	38.00	1.03 × 10^3^	Non-toxic
13	K_2_Cr_2_O_7_	10.00	20.00	80.00	96.66	100.00	38.00	23.08	Highly toxic

Each abbreviation represents plant and collection site within the study area while K_2_Cr_2_O_7_ was the standard used during the experiment (positive control); VA—*Vernonia amygdalina*, OG—*Ocimum gratissimum*, OAB—Oria Abraka, U.A.B.—Urhovie Abraka, A.A.B.—Ajalom Abraka, E.G.K.—Egbo Kokori, K—Kokori, SK—Samagidi Kokori. # — Inference based on modified Meyer classification of crude extracts and pure substances into generally toxic (LC_50_ value < 1000 µg/mL) and non-toxic (LC_50_ value > 1000 µg/mL); the lower the LC_50_ value, the more toxic it is to test organisms, in particular, values less than <100 µg/mL are considered highly toxic.

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
