# Peer review of "Heavy Metals, Proximate Analysis and Brine Shrimp Lethality of Vernonia amygdalina and Ocimum gratissimum Growing in Crude Oil-Rich Delta State, Nigeria"

_foods, 2021, doi:10.3390/foods10122913_

Round 1

Reviewer 1 Report

Introduction must be improved. Heavy metals' standard limits set by FAO/WHO must be given for each metals. 

Materials and methods must be improved. If the plants used for analysis are cultivated, agricultural practices (chemical fertilization, irrigation, pesticides used etc.) and detailed soil analysisi for each region must be given in details. 

Discussion part is poor. More literature must be given and discussed former scientific data. 

Reviewer 2 Report

The authors did not a good work from an experimental point of view and I recommend major revisions.

More specific:

L13: Latin names in italics.

L50: You have a different font size.

L52, 68: Why in italics? The same as above.

L94: Replace the image Table 1 with normal Table.

L97: Appear on the map of Nigeria with color the location of the large map.

L119: Replace the special symbol ‘’g’’ with the normal ‘’g’’.

L146: How many calibration points did you use from each metal?

L154: Replace the image Table 2 with normal Table.

L174: What test was performed in ANOVA?

L195: Replace the image Table 3 with normal Table.

L285: Replace the image Table 4 with normal Table.

Round 2

Reviewer 2 Report

The paper has been revised according to the suggestions and criticisms of the reviewers. In this revised version, the paper has improved its quality.